# Low Density Lipoprotein Cholesterol Decreases the Expression of Adenosine A_2A_ Receptor and Lipid Rafts-Protein Flotillin-1: Insights on Cardiovascular Risk of Hypercholesterolemia

**DOI:** 10.3390/cells13060488

**Published:** 2024-03-11

**Authors:** Marie-Charlotte Chaptal, Marie Maraninchi, Giorgia Musto, Julien Mancini, Hedi Chtioui, Janine Dupont-Roussel, Marion Marlinge, Julien Fromonot, Nathalie Lalevee, Florian Mourre, Sophie Beliard, Régis Guieu, René Valero, Giovanna Mottola

**Affiliations:** 1Centre de Recherche en Cardiovasculaire et Nutrition (C2VN), Aix-Marseille Université, INSERM 1263, INRAE 1260, 13005 Marseille, France; marie-charlotte.chaptal@univ-amu.fr (M.-C.C.); marie.maraninchi@univ-amu.fr (M.M.); giorgia.musto@unina.it (G.M.); julien.mancini@univ-amu.fr (J.M.); julien.fromonot@univ-amu.fr (J.F.); nathalie.lalevee@univ-amu.fr (N.L.); florian.mourre@ap-hm.fr (F.M.); sophie.beliard@ap-hm.fr (S.B.); regis.guieu@univ-amu.fr (R.G.); rene.valero@ap-hm.fr (R.V.); 2Department of Pharmacy, University of Naples Federico II, 80131 Naples, Italy; 3Department of Nutrition, Metabolic Diseases and Endocrinology, Hospital La Conception, APHM, 13005 Marseille, France; hedi.chtioui@ap-hm.fr (H.C.); jeanine.dupont@univ-amu.fr (J.D.-R.); 4Secteur de Biochimie, Biogenopôle, Hôpital de la Timone, APHM, 13005 Marseille, France

**Keywords:** hypercholesterolemia, LDL-C, cholesterol, adenosine receptor, lipid rafts, Flotillin-1, cardiovascular risk

## Abstract

High blood levels of low-density lipoprotein (LDL)-cholesterol (LDL-C) are associated with atherosclerosis, mainly by promoting foam cell accumulation in vessels. As cholesterol is an essential component of cell plasma membranes and a regulator of several signaling pathways, LDL-C excess may have wider cardiovascular toxicity. We examined, in untreated hypercholesterolemia (HC) patients, selected regardless of the cause of LDL-C accumulation, and in healthy participants (HP), the expression of the adenosine A_2A_ receptor (A_2A_R), an anti-inflammatory and vasodilatory protein with cholesterol-dependent modulation, and Flotillin-1, protein marker of cholesterol-enriched plasma membrane domains. Blood cardiovascular risk and inflammatory biomarkers were measured. A_2A_R and Flotillin-1 expression in peripheral blood mononuclear cells (PBMC) was lower in patients compared to HP and negatively correlated to LDL-C blood levels. No other differences were observed between the two groups apart from transferrin and ferritin concentrations. A_2A_R and Flotillin-1 proteins levels were positively correlated in the whole study population. Incubation of HP PBMCs with LDL-C caused a similar reduction in A_2A_R and Flotillin-1 expression. We suggest that LDL-C affects A_2A_R expression by impacting cholesterol-enriched membrane microdomains. Our results provide new insights into the molecular mechanisms underlying cholesterol toxicity, and may have important clinical implication for assessment and treatment of cardiovascular risk in HC.

## 1. Introduction

Systemic cholesterol excess, i.e., hypercholesterolemia (HC), defined by high blood levels of total-cholesterol (TC) and low-density lipoprotein (LDL) cholesterol (LDL-C), has been associated with atherosclerosis and coronary artery disease (CAD) due to promotion of the accumulation of pathological macrophage (foam cells) within arterial walls. However, because cholesterol is a component of all mammalian cell membranes, excess levels might have a wider impact on tissue and organ functionality and multiple mechanisms might explain cholesterol toxicity and the increased cardiovascular risk associated with hypercholesterolemia.

Atherosclerosis is the major cause of cardiovascular disease and stroke [1]. This slowly progressing pathology, characterized by activation of inflammation and endothelial dysfunction, is in large part initiated by LDL and triglycerides (TG)-rich lipoprotein accumulation in the intima of vessels. Lipids are oxidized and internalized by recruited macrophages, which then transform into foam cells, accumulate within the arterial walls, and initiate lesions. Lesion growth can reduce blood flow and may cause angina, particularly during exercise. Lesions may also be unstable, and their rupture can create a local clot that may completely obstruct blood flow and lead to a heart attack or a stroke. High LDL-C blood levels have been associated with an increased risk of lesion formation, while high-density lipoprotein cholesterol (HDL-C) levels are generally considered protective, because they are involved in the recycling and removal of excess cholesterol [2,3]. Hypercholesterolemia may be idiopathic, secondary to another disease or condition, multifactorial or genetic, such as in familial hypercholesterolemia (FH), which is mainly associated with mutations in genes encoding proteins involved in LDL transport (apolipoprotein B, apoB), or internalization [LDL receptor (LDLR) and Proprotein Convertase Subtilisin/Kexin Type 9 (PCKS9)] [4,5]. To date, the majority of hypercholesterolemia studies have focused on the link between LDL-C, foam cells and lesion formation. However, cholesterol is an essential molecule in mammalian metabolism. While it is a precursor for bile acid and steroid hormone biosynthesis, it is principally an indispensable component of all mammalian cells, particularly of the plasma membrane, where it constitutes 25% of total lipids. By modulating membrane physico-chemical properties, it controls membrane permeability, fluidity, and structural and functional organization [6,7,8]. Together with glycosphingolipids, cholesterol forms membrane micro-domains, known as lipid rafts. These microdomains are associated with specific proteins, including Flotillins, and regulate intracellular trafficking and signaling of several plasma membrane receptors [9,10,11]. Almost all mammalian cells can synthesize cholesterol and internalize LDL-C via LDLR. Therefore, perturbation in cholesterol homeostasis and LDL-C levels would be expected to have a greater impact on human metabolism [8].

Adenosine A_2A_ receptor (A_2A_R) is a G-protein-coupled receptor expressed mainly in immune cells and platelets, as well as in vascular smooth muscle and endothelial cells [12,13,14,15]. It activates intracellular processes through G-proteins in response to adenosine binding. A_2A_R has a protective immunosuppressive function: it attenuates production of pro-inflammatory cytokines and promotes synthesis of anti-inflammatory molecules. A_2A_R activation also induces relaxation of vascular smooth muscle, coronary and peripheral artery vasodilation, and regulates cardiac rhythm [16,17,18,19,20]. Alterations in A_2A_R expression and function have been associated with CAD. A decrease in basal A_2A_ receptor expression has been found in patients with CAD, or subjects with a positive stress test [21,22,23] and a reduced fractional flow reserve (FFR), which has led to the analysis of A_2A_R expression and function being proposed as a prognostic biomarker of CAD severity [21]. In recent years, a reciprocal link has been described between cholesterol metabolism and A_2A_R [24,25,26,27,28,29,30,31,32]. A_2A_R appears to modulate intracellular cholesterol metabolism at several stages: synthesis, efflux, influx, and intracellular transport [28,29,30,31,32,33]. On the other hand, specific cholesterol-binding sites have been identified in the structure of A_2A_R [24,25,26,27] and A_2A_R has been proposed associated with lipid rafts [33], suggesting that cholesterol is an important regulator of A_2A_R function. Therefore, we hypothesize that alterations in A_2A_R expression and function may be present in hypercholesterolemia.

We have previously shown, in a pilot study of a small cohort of FH patients, where LDL-C excess is genetically induced, that high LDL-C was linked to decreased A_2A_R expression in PBMCs [34]. In the present study, we analyzed the relationship between elevated blood LDL-C levels, intracellular cholesterol levels and A_2A_R expression in a larger cohort of healthy participants (HP) and HC patients, chosen regardless the origin of their LDL-C excess and without any hypolipidemic treatment, which are known to have pleiotropic effects independent of their cholesterol-lowering properties [35,36]. To investigate whether alterations in A_2A_R expression might be explained by alterations in membrane micro-domains, we examined the expression of the lipid raft marker Flotillin-1. Interestingly, we found significant differences between the two groups of individuals which we believe provide new insight into the cellular mechanisms of cardiovascular risk in hypercholesterolemia.

## 2. Materials and Methods

### 2.1. Study Population

This protocol was a prospective, comparative, single-center observational and experimental study. Patients were recruited from the Nutrition, Metabolic Diseases and Endocrinology Department of the Hospital “La Conception”, APHM, (Marseille, France) and met the following inclusion criteria: age (20–65 years), high TC (>199.9 mg/dL or >5.17 mM) and LDL-C (>160 mg/dL or >4.14 mM) blood levels, and no hypolipidemic treatment.

Healthy participants were recruited at the Service of Biochemistry, Biogenopôle, Hospital La Timone (Marseille, France) and met the following inclusion criteria: TC (100–200 mg/dL or 2.59–5.17 mM) and LDL-C (59.9–160 mg/dL or 1.55–4.14 mM) blood levels, no hypolipidemic treatment, lack of history of diabetes, hypertension, previously elevated cholesterol, or hypertriglyceridemia. The two groups were matched according to age, gender, and body mass index (BMI). The main difference between the two groups was in blood levels of LDL-C.

All biochemical parameters were analyzed by the Biochemistry Department, Medical Biology Laboratory (LBM), Hospital La Timone, APHM (Marseille, France). This study was conducted in accordance with the Declaration of Helsinki on experiments involving humans, and approved by the institutional Ethics Committee (CPP Sud-Méditerranée II, Marseille, France), 21 June 2021, number ID-RCB: 2021-A01196-35). All participants gave written informed consent.

### 2.2. Peripheral Blood Mononuclear Cell (PBMC) Isolation

Blood samples were collected by venipuncture from the brachial vein into 8 mL tubes containing sodium citrate/Ficoll (BD Vacutainer CPT, Becton Dickinson, Franklin lakes, NJ, USA). PBMCs were prepared according to the manufacturer’s instructions. Briefly, blood samples were centrifuged for 30 min at 2000 RCF at 20 °C. PBMCs were then collected from the plasma/Ficoll interface and washed twice with phosphate-buffered saline (PBS). After cell counting, aliquots were used for SDS-PAGE and Western blot analysis and the remainder frozen in freezing medium containing 10% DMSO (C6164, Sigma^®^, Saint-Quentin-Fallavier Cedex, France) at −80 °C.

### 2.3. PBMCs Treatment with LDL-C Enriched Medium

PBMCs isolated from a healthy participant were incubated with or without a high LDL-C-enriched human serum, ref. 360-10, LeeBiosolutions, Maryland Heights, MO, USA) medium at a final LDL-C concentration of 309 mg/dL (or 8 mM), at 37 °C for 6, 24 and 48 h. Cells were then centrifuged and used for SDS-PAGE and Western blot analysis of A_2A_R and Flotillin-1 expression.

### 2.4. SDS-PAGE and Western Blot Analysis

A_2A_R and Flotillin-1 expression in PBMCs were determined by SDS-PAGE and Western blot analysis as previously described [37]. PBMC pellets were homogenized in RIPA buffer (1% Triton X-100, 0.5% sodium deoxycholate, 0.1% SDS, 50 mM Tris pH 7.4, 150 mM NaCl, 0.5 mM EDTA). Protein concentrations were measured using a BCA Protein Assay Kit (Novagen^®^, Gauteng, South Africa) according to the manufacturer’s instructions. For each sample, 10 µg of total proteins were precipitated by addition of nine volumes of 100% cold acetone and then incubated 2 h at 4 °C. After centrifugation for 10 min at 14,000 RCF supernatants were discarded, and pellets dried at room temperature. Total protein pellets were solubilized using Laëmmli buffer containing 5% ß-mercaptoethanol and TBS 1× and then heated 15 min at 65 °C. Samples were then submitted to standard 12% polyacrylamide gel electrophoresis under reducing conditions, before transfer to a polyvinylidene difluoride membrane. The membrane was incubated with Adonis, a home-made antibody (9 µM stock solution, diluted 1/3500 [37]), or human Flotillin-1 antibody (0.55 mg/mL stock solution, diluted 1/10,000, ab133497, Abcam^®^, Cambridge, UK). Blots were developed using alkaline phosphatase-labeled anti-mouse (A2179, Sigma^®^) and anti-rabbit secondary antibodies (A3137, Sigma^®^), diluted at 1/3500, using the colorimetric substrate BCIP/NBT (Sigma^®^). The upper part of each gel was stained using amido black solution (0.05% amido black in a solution of 10% methanol/10% acetic acid) to quantify the total amount of proteins loaded in each lane. For quantification, the intensity of the band corresponding to each protein (45 kDa for A_2A_R or 47 kDa for Flotillin-1) or of total proteins was measured by densitometry using Image J software (1.53k, Wayne Rasband and contributors, National Institutes of Health, Stapleton, NY, USA). Results were expressed in arbitrary units (A.U.) as the ratio between the intensity of adenosine A_2A_ receptor or Flotillin-1 bands and the intensity of the stained total protein band. For each subject (patient or healthy donor), the A.U. value is the average of a minimum of *n* = 3 repeated Western blot experiments.

### 2.5. Statistical Analysis

Categorical parameters are expressed as counts (%) and continuous parameters as mean ± standard deviation (SD). When comparing patients to HP, the categorical parameters were analyzed by Χ^2^ or Fisher’s exact tests. Continuous variables were analyzed using *t*-tests with Welch’s correction when variance differed. Associations between continuous parameters were estimated using Pearson correlation coefficients (r). All tests were two-sided and statistical significance was defined as *p* ≤ 0.05 *; *p* ≤ 0.001 **; *p* ≤ 0.0001 ***. Statistical analyses were performed using IBM SPSS Statistics 27.0 (IBM Inc., New York, NY, USA).

## 3. Results

### 3.1. Hypercholesterolemic Patient and Healthy Subject Characteristics

The demographic and clinical characteristics of the participants are shown in Table 1. We recruited 37 HC patients and 31 HP. The two groups were similar in terms of age, gender, and body mass index (BMI): HC patients (70.96% women, 29.03% men; age 46.03 ± 15.43 years; BMI 23.89 ± 3.80 kg/m^2^) and HP (67.57% women, 32.43% men; age 40.71 ± 13.15 years; BMI 23.00 ± 3.73 kg/m^2^) (Table 1). As expected, mean TC levels were significantly higher in the HC patient group (8.06 ± 1.40 mM) than in the HP group (4.752 ± 0.642 mM) (*p* < 0.0001). A similar difference was observed for LDL-C levels: 5.90 ± 1.29 mM for HC patients vs. 2.683 ± 0.564 mM for HP (*p* < 0.0001). In line with the LDL-C variation, the two groups also had a significant difference in apoB concentration (1.51 ± 0.27 mM HC vs. 0.778 ± 0.144 mM HP, *p* < 0.0001). No significant difference was observed for HDL-cholesterol (HDL-C, *p* = 0.807) and TG (*p* = 0.095), indicating that the difference in TC concentration is associated with the difference in LDL-C concentrations. The two groups showed no significant differences regarding lipoprotéine A (Lp(a)) and homocysteine (Hcy) concentrations (*p* = 0.509 and *p* = 0.953, respectively), two factors that have been associated with cardiovascular risk [38]. Additionally, no significant difference was found for biomarkers of systemic inflammatory response, including C-reactive protein (CRP) (*p* = 0.251), haptoglobin (*p* = 0.291), orosomucoid (*p* = 0.361), prealbumin (*p* = 0.920) and albumin (*p* = 0.069). Significant differences were observed only in the case of transferrin and ferritin, two modulators of iron metabolism: transferrin levels were lower (2.27 ± 0.32 g/L vs. 2.62 ± 0.49 g/L, *p* < 0.001) and ferritin blood levels were higher (102.04 ± 110.45 µg/L and 57.81 ± 56.11 µg/L, *p* < 0.05) in HC patients compared to HP.

### 3.2. Analysis of A_2A_ Receptor Expression in Hypercholesterolemic Patients and Healthy Participants

Expression of A_2A_R in PBMCs of hypercholesterolemia patients and HP was analyzed, quantified, and found to be significantly lower (46% lower) in HC patients (P) compared to HP [0.124 ± 0.064 arbitrary units (A.U.) vs. 0.229 ± 0.092 A.U., *p* < 0.0001] (Figure 1A,B). Additionally, we observed that A_2A_R expression was negatively correlated to both TC (r = −0.442, *p* = 0.001) and LDL-C (r = −0.459, *p* < 0.0001) blood levels (Figure 1C).

### 3.3. Analysis of Flotillin-1 Expression in Hypercholesterolemic Patients and Healthy Participants

To investigate the possible mechanism of the decreased expression of A_2A_R, we analyzed, in parallel, the expression of Flotillin-1, a protein associated with lipid raft microdomains in the plasma membrane [39], whose localization is correlated with intracellular cholesterol distribution [40]. We found that Flotillin-1 expression was significantly lower (53% less) in HC patients compared to HP (0.105 ± 0.063 vs. 0.198 ± 0.117 A.U., *p* < 0.0001) (Figure 2A,B). Moreover, as seen for A_2A_R, a negative correlation was seen between Flotillin-1 expression and TC (r = −0.456, *p* < 0.0001) and LDL-C (Pearson r = −0.449, *p* < 0.0001) (Figure 2C). Interestingly, A_2A_R expression was also positively correlated with Flotillin-1 expression levels (r = 0.404, *p* = 0.001) (Figure 3).

### 3.4. Effect of LDL-C Supplementation on A_2A_R and Flotillin-1 Expression

To investigate whether the effect of high LDL-C levels is a direct cellular response, we incubated healthy individual PBMCs with or without highly LDL-C-enriched (8 mM final concentration) medium. As shown in Figure 4, A_2A_R expression decreased rapidly and significantly over time in the presence of LDL-C compared to control (without LDL-C-enrichment) (Time 6 h: 0.058 ± 0.006 (+) vs. 0.344 ± 0.165 (−) A.U., *p* = 0.003; T24 h: 0.050 ± 0.094 (+) vs. 0.313 ± 0.132 (−) A.U., *p* = 0.006; T48 h: 0.023 ± 0.033 (+) vs. 0.249 ± 0.173 (−) A.U., *p* = 0.020; n = 3) (Figure 4A). Moreover, a decrease in the Flotillin-1 47 kDa band was also observed at these time points (T6 h: 0.082 ± 0.039 (+) vs. 0.210 ± 0.070 (−) A.U., *p* = 0.078; T24 h: 0.108 ± 0.021 (+) vs. 0.279 ± 0.092 (−) A.U., *p* = 0.017; T48 h: 0.097 ± 0.061 (+) vs. 0.285 ± 0.064 (−) A.U., *p* = 0.009).

## 4. Discussion

In this study, we describe the specific association between high LDL-C blood levels and decreased adenosine A_2A_ receptor expression in PBMCs of untreated hypercholesterolemia patients, together with the negative correlation between A_2A_R expression and TC and LDL-C blood levels. Furthermore, we report similar alterations and correlations for Flotillin-1, a protein marker of cholesterol-enriched membrane microdomains. Changes in A_2A_R as well as Flotillin-1 were found to be directly induced in healthy PBMCs by LDL-C enrichment of cell culture medium. Interestingly, A_2A_R was also positively correlated with Flotillin-1 expression in PBMCs.

Taken together, these data suggest that alterations in A_2A_R expression are associated and correlated with high blood levels of LDL-C and hypercholesterolemia, regardless of the cause of LDL-C excess. The decrease in Flotillin-1 expression in HC patients and its correlation to cholesterol levels suggest a perturbation of intracellular cholesterol levels and lipid raft organization in HC patients and may explain the tight correlation of A_2A_R expression with cholesterol levels in hypercholesterolemia.

In recent years, a reciprocal relationship has been described between cholesterol metabolism and A_2A_ adenosine receptors. Cholesterol appears to be an important regulator of A_2A_R function since several X-ray crystallographic and simulation models have suggested the presence of up to three cholesterol interaction sites in the molecular structure of A_2A_R [10,41,42]. Cholesterol depletion in rat embryonic cortical neurons [43] and in erythrocytes [44] by cyclodextrin abolishes A_2A_R activation of cyclic adenosine 3′,5′-monophosphate (cAMP) synthesis by the agonist CGS21680. Gs proteins and adenylyl cyclase are also present in cholesterol-enriched microdomains (lipid rafts) in the plasma membrane in various cell types [6,45,46] and it has been proposed that A_2A_R associates with these domains [47]. Additionally, A_2A_Rs appears to participate in the regulation of cholesterol metabolism at several stages: synthesis, efflux, influx, and intracellular transport [28,29,30,31,32]. Patients with dyslipidemia showed a reduction in myocardial blood flow and reduced coronary artery dilation upon intravenous administration of adenosine and, because A_2A_R is implicated in coronary artery dilation, it has been suggested that its expression or function may be impacted in dyslipidemia [48]. In the present study, our results confirm and extend our previous observations on Familial hypercholesterolemia patients [34] that a clear relationship at the systemic level exists between cholesterol metabolism and A_2A_ adenosine receptor expression. Indeed, hypercholesterolemic patients, selected independently of the origin of LDL-C accumulation, show decreased expression of A_2A_R in PBMCs. No other difference was noted between the two groups in terms of age, gender, blood lipid markers (HDL-C or triglycerides), obesity (BMI), history of diabetes or hypertension, in terms of the principal risk factors for hypercholesterolemia. Interestingly, the higher LDL-C blood levels were, the lower A_2A_R expression was, and this downregulation might be directly induced by LDL-C, as suggested by our in vitro experiments on healthy PBMCs. Our results thus suggest that, regardless of the cause of LDL-C accumulation (primary or secondary), this excess is sufficient to lower A_2A_R expression.

To date, studies on hypercholesterolemia and atherosclerosis have mostly focused on mechanisms of LDL-C oxidation and internalization by recruited macrophages, which then transform into foam cells, accumulate within arterial walls, and initiate lesion formation [1]. Nonetheless, because cholesterol is an indispensable ubiquitous lipid, LDL-R is expressed in almost all mammalian cells and LDL-C internalization is tightly regulated. Thus, it could be expected that in a context of perturbations in cholesterol metabolism, i.e., LDL-C accumulation, every cell could potentially recognize and respond to this alteration. However, little is currently known regarding intracellular cholesterol levels and cholesterol distribution being modified in PBMCs or other cells exposed to high LDL-C concentrations and the mechanism underlying such modifications. Pfisterer et al. (2022) recently quantified LDL uptake and storage in cytoplasmic droplets in PBMCs of hypercholesterolemic patients [49]. They reported that increased circulating LDL is associated with low intracellular LDL-C uptake and low lipid storage in lipid droplets, which suggests that intracellular membrane cholesterol levels are also affected in PBMCs. To investigate whether, and how, in the presence of LDL-C excess, intracellular membrane cholesterol levels are altered and impact A_2A_R expression in our study, we evaluated the expression of Flotillin-1, a marker of cholesterol-enriched intracellular membrane domains. Flotillin-1 is essential for Niemann-Pick C1-like 1 (NPC1L1)-mediated cellular cholesterol uptake and, together with NPC1L1, forms cholesterol-enriched membrane microdomains in the plasma membrane [40]. Cholesterol recruitment into low-density membrane fractions is substantially reduced in cells with reduced levels of Flotillin-1. To our knowledge, our study is the first to report that Flotillin-1 expression is decreased in PBMCs from hypercholesterolemic patients, and this decrease is induced in healthy PBMCs after LDL-C supplementation. This downregulation might reflect an alteration in lipid rafts composition and/or function at the plasma membrane, which may destabilize A_2A_R causing its degradation. The significant positive correlation observed between Flotilin-1 and A_2A_R protein expression supports the hypothesis of A_2A_R localization to Flotillin 1-positive and cholesterol enriched membrane microdomains. Nonetheless, future experiments are needed to elucidate whether and how association to lipid rafts might modulate A_2A_R localization, stability, and function and the impact of LDL-C excess on such modulation.

The identification of altered Flotillin-1 and A_2A_R expression in hypercholesterolemia provides new insights into the mechanisms of LDL-C excess toxicity. Aside from the formation and accumulation of foam cells in the arterial wall, multiple additional mechanisms might explain the cardiovascular risk associated with high blood LDL-C levels:(1)A_2A_R is activated in response to hypoxia and ischemia and has a protective function, by attenuating the production of pro-inflammatory cytokines such as interleukins IL-2 and IL-4, TNF-α and INF-γ, and by promoting the synthesis of anti-inflammatory molecules such as interleukin IL-10 [50,51,52,53,54]. Analysis of A_2A_R expression and function in PBMCs are also similar to expression in the myocardium [55], coronary arteries, aorta [22] and iliac arteries [23], these tissue sites being the preferred location for cholesterol deposits. A_2A_R activation induces relaxation of vascular smooth muscle cells, resulting in coronary and peripheral vasodilation [16,20,56,57,58]. Therefore, decreased expression of A_2A_R in PBMCs could potentially decrease adenosine-induced vasodilation in response to ischemia and therefore prevent the compensatory mechanisms that are activated in response to obstruction of blood flow due to lesion formation.(2)Because Flotillin-1 is a ubiquitous component of cholesterol-enriched membrane domains [59,60] (REF, alteration in its expression might indicate a destabilization of these membrane domains, which might have a larger impact on the localization and function of other plasma membrane-associated proteins in PBMCs, as well as in other cell types exposed to high LDL levels). Lipid rafts are already known as crucial modulators of receptor-mediated signaling processes [61,62]. The results of this report highlight the relevance to broadly investigate the effect of LDL-C excess on lipid rafts organization and function and to screen on all signaling pathways that might be modified. This knowledge might contribute to understanding the molecular mechanisms underlining LDL-C toxicity and to find novel targets and therapeutic approaches against hypercholesterolemia, as for patients resistant to current hypolipidemic treatments.

Potential correlations of decreased expression of A_2A_R and Flotillin-1 with altered expression of common systemic inflammatory biomarkers or of other factors associated with cardiovascular risk, such as homocysteine [63,64] and lipoprotein(a) [38], were also looked for. We observed no difference between the HC patient and HP groups with the exception of two important biomarkers of iron metabolism: transferrin, a protein regulating iron transport, and ferritin the protein involved in iron storage. Nonetheless, serum transferrin and ferritin are acute phase factors. Lower transferrin (<2 g/L or 200 mg/dL) [65] and higher ferritin (>300 μg/L serum ferritin in men and >200 μg/L in women [66], blood concentrations are recognized as biomarkers in inflammation and might even contribute to atherosclerosis [67] and CVD [68]. Abnormal iron metabolism leads to ferroptosis, a regulated cell death that is characterized by the accumulation of lipid peroxidation products [69,70], which has been associated with cardiac inflammation and dysfunction [71,72,73]. While still in the normal range, we found significantly lower transferrin levels and higher ferritin levels in HC patients compared to healthy individuals (Table 1), which might indicate the beginning of an inflammatory processus. Interestingly, the iron exporter, ferroportin, in macrophages has been mostly detected in detergent-resistant membranes containing Flotillin-1 [74], suggesting a possible link. Further investigation will be required to understand whether in our cohort inflammation and dysfunction are in progress, i.e., quantification of pro-inflammatory and anti-inflammatory cytokines and of endothelial and cardiac dysfunction biomarkers [75]. Moreover, a long-term follow up might help to understand whether these variations over time will evolve and be associated with cardiovascular complications in our cohort. In such case, the identification of A_2A_R and Flotillin-1 downregulation and transferrin and ferritin alterations might represent a novel panel of very early predictive biomarkers of the cardiovascular risk in hypercholesterolemia. In addition, our study also presents new perspectives on the use of A_2A_R modulators to counteract the cardiovascular risk of LDL-C excess. In fact, caffeine, which induces A_2A_R overexpression [76], has been shown to protect against mortality in cardiovascular disease [77] and counteract the cardiovascular deleterious effects associated with HC [78].

## 5. Conclusions

In this study, we provide insights on novel molecular mechanisms that could explain LDL-C cell toxicity: a decreased expression of adenosine A_2A_ receptor is likely associated with decreased anti-inflammatory responses and vessels vasodilation, and thus may contribute to increasing cardiovascular risk associated with hypercholesterolemia. Moreover, the decrease in the lipid raft-associated protein Flotillin-1 suggests a broader impact of LDL-C excess on the organization and functionality of cholesterol-enriched membrane domains, which may perturb other signaling pathways. A better understanding of these mechanisms and the discovery of novel key factors will contribute to improve therapeutic options for the treatment of hypercholesterolemia, particularly in case of resistance or development of pathological complications to the current hypolipidemic treatments. This study also highlights the possible employment of novel diagnostic biomarkers of inflammation and progression towards atherosclerosis and cardiovascular pathology in hypercholesterolemia. The evaluation of A_2A_R and Flotillin-1 protein expression, together with the quantification of transferrin and ferritin blood levels, in future might be integrated in the tools for cardiovascular risk assessment in HC patients.

## Figures and Tables

**Figure 1 cells-13-00488-f001:**
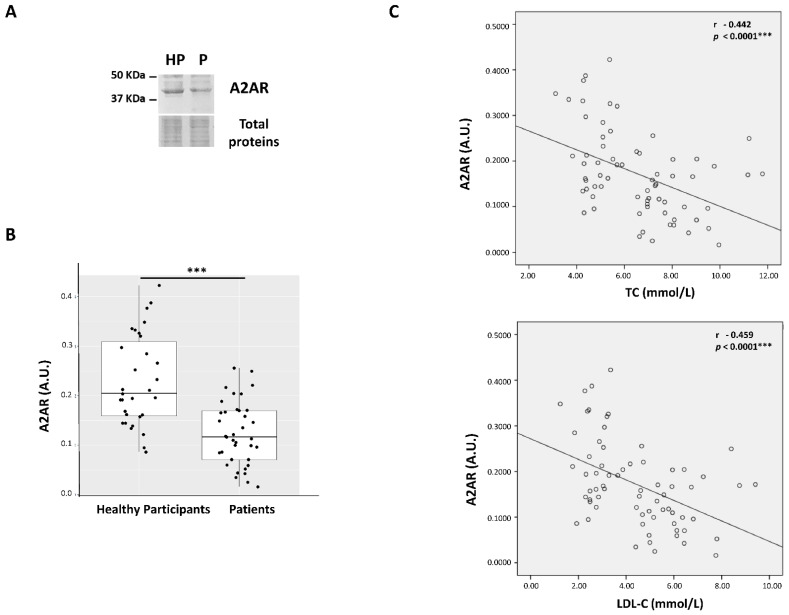
A_2A_ receptor expression is found to be lower in hypercholesterolemic patients compared to healthy participants and negatively correlates to TC or LDL-C blood levels. Representative images of the A_2A_R 45 kDa band visualized by Western blot in PBMCs of HP or hypercholesterolemic patients (P) (**A**). After Image J densitometry analysis, A_2A_R expression was expressed in arbitrary units (A.U.) as the ratio of the intensities between the A_2A_R 45k Da band and the stained total protein band. Each individual value represents the mean ± SD of three separate experiments. *t*-test was used for comparisons between groups (*p* < 0.0001 ***) (**B**). A_2A_R expression levels (in A.U.) of the entire group (hypercholesterolemic patients and HP) were correlated to TC or LDL-C blood levels. Pearson’s correlation coefficient and the coefficient of determination r^2^ were calculated, *p* ≤ 0.05 was significant (**C**).

**Figure 2 cells-13-00488-f002:**
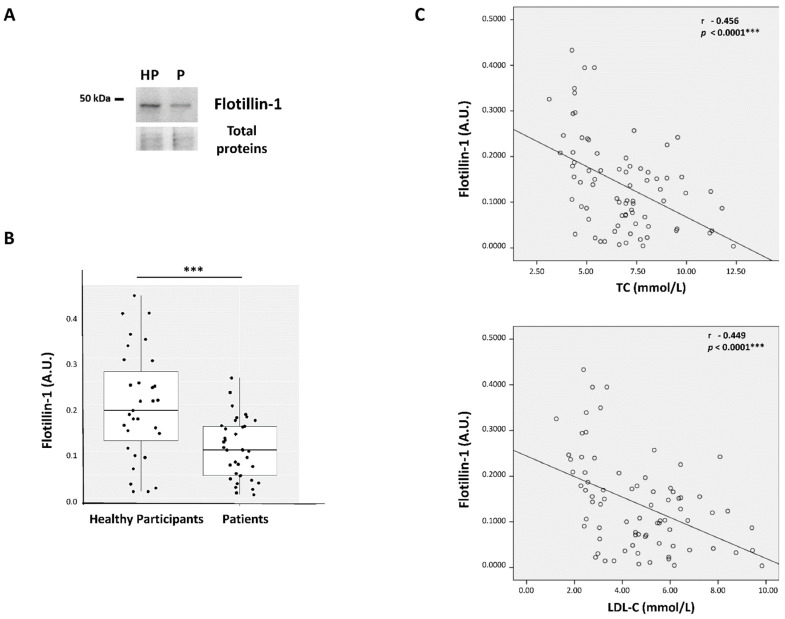
Flotillin-1 expression is found to be lower in hypercholesterolemic patients compared to healthy participants and negatively correlates to TC or LDL-C blood levels. Representative image of the Flotillin-1 47k Da band visualized by Western blot in PBMCs from HP or hypercholesterolemic patients (P) (**A**). After Image J densitometry analysis, Flotillin-1 expression was expressed in arbitrary units (A.U.) as the ratio of the intensities between the Flotillin-1 47 kDa band and the stained total protein band. Individual A.U. values represent the mean ± SD from three separate experiments. *t*-tests were used for comparisons between groups (*p* < 0.0001 ***) (**B**). Flotillin-1 expression levels (in A.U.) for the whole subject group (hypercholesterolemic patients and HP) were correlated to TC or LDL-C blood levels. Pearson’s correlation coefficient and the coefficient of determination r^2^ were calculated, *p* ≤ 0.05 was considered significant (**C**).

**Figure 3 cells-13-00488-f003:**
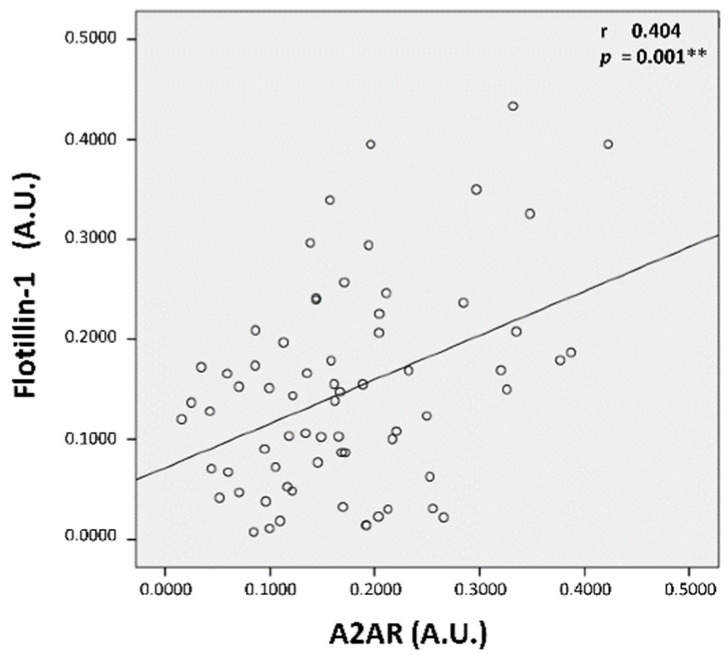
Positive correlation between A_2A_R and Flotillin-1 expression in all individuals. A_2A_R and Flotillin-1 expression levels (in A.U.) were positively correlated for the whole participant group. Pearson’s r coefficient of correlation and the coefficient of determination r^2^ were used for the correlation study, *p* ≤ 0.001 considered significant (**).

**Figure 4 cells-13-00488-f004:**
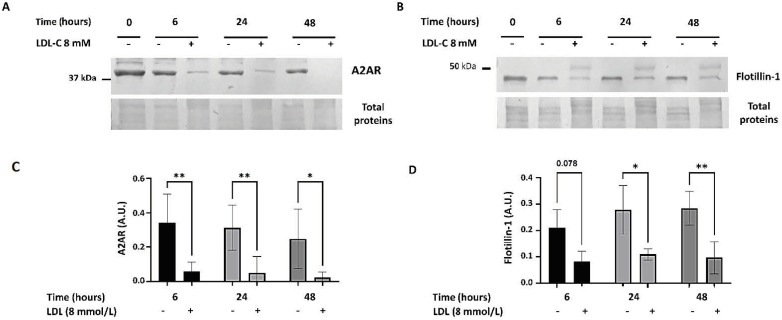
A_2A_R and Flotillin-1 expression decreases in PBMCs after incubation with LDL-C enriched (8 mM) medium. PBMCs (T0) were incubated with (+) or without (−) LDL-C enriched (8 mM final concentration) medium for 6, 24 and 48 h. Representative image of the A_2A_R 45 kDa band (**A**) and the Flotillin-1 47 kDa band (**B**) visualized by Western blot. After Western blotting and Image J densitometry analysis, A_2A_R and Flotillin-1 expression is shown in arbitrary units (A.U.) as the ratio of the intensities between the A_2A_R or Flotillin-1 band and the stained total protein band. Data are represented as mean ± SD from five separate experiments (for A_2A_R) and three experiments (Flotillin-1). A 2-way ANOVA test was performed, *p* ≤ 0.05 *; *p* ≤ 0.001 ** (**C**,**D**).

**Table 1 cells-13-00488-t001:** Biological characteristics of the healthy participants and patients’ groups.

	Healthy Participants (31)	Patients (37)	*p*
**Gender**	22 women; 9 men	25 women; 12 men	0.762
**Age**	40.71 ± 13.15	46.03 ± 15.43	0.135
**BMI (kg/m^2^)**	23.00 ± 3.73	23.89 ± 3.80	0.345
**TC (mM)**	4.75 ± 0.64	8.06 ± 1.40	<0.0001 ***
**HDL-C (mM)**	1.62 ± 0.42	1.60 ± 0.35	0.807
**LDL-C (mM)**	2.68 ± 0.56	5.90 ± 1.29	<0.0001 ***
**TG (mM)**	1.02 ± 0.48	1.24 ± 0.55	0.095
**apoB (mM)**	0.78 ± 0.14	1.51 ± 0.27	<0.0001 ***
**Lp(a) (g/L)**	0.24 ± 0.22	0.29 ± 0.32	0.509
**CRP (mg/L)**	1.37 ± 2.29	2.29 ± 3.87	0.251
**Hcy (µM)**	11.28 ± 3.72	11.34 ± 3.64	0.953
**Neutrophils (N) (10^9^/L)**	3.87 ± 1.33	3.77 ± 1.82	0.803
**Lymphocytes (L) (10^9^/L)**	2.06 ± 0.36	1.86 ± 0.61	0.180
**NLR (N/L)**	1.96 ± 0.71	2.22 ± 1.28	0.328
**Albumin (g/L)**	42.59 ± 2.61	43.97 ± 3.15	0.069
**Transferrin (g/L)**	2.62 ± 0.49	2.27 ± 0.32	0.001 **
**Iron (µM)**	18.53 ± 9.88	15.60 ± 5.00	0.168
**FCT (µM)**	65.48 ± 12.23	56.97 ± 8.09	0.002 *
**TSC (%)**	27.30 ± 13.44	28.61 ± 7.70	0.656
**Ferritin (µg/L)**	57.81 ± 56.11	102.04 ± 110.45	0.045 *
**Haptoglobin (g/L)**	1.04 ± 0.35	1.16 ± 0.50	0.291
**Orosomucoid (g/L)**	0.62 ± 0.17	0.67 ± 0.21	0.361
**Prealbumin (g/L)**	0.28 ± 0.05	0.28 ± 0.05	0.920

Data are mean ± SD. Mean dates were compared between patients with hypercholesterolemia (HC) and healthy participants (HP). *p* ≤ 0.0001 ***, *p* ≤ 0.001 **, *p* ≤ 0.05 * are significant. BMI, body mass index, kg/m^2^; TC, total cholesterol; HDL-C, HDL-cholesterol; LDL-C, LDL-cholesterol; TG, triglycerides; apoB, Apoliprotein B; Lp(a), lipoprotein(a); CRP, C-reactive protein; Hcy, homocystein; NLR, neutrophils-lymphocytes ratio; FCT, iron fixation capacity of transferrin; TSC, transferrin saturation coefficient.

## Data Availability

Data are contained within the article.

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
