# Peer review of "Low Density Lipoprotein Cholesterol Decreases the Expression of Adenosine A2A Receptor and Lipid Rafts-Protein Flotillin-1: Insights on Cardiovascular Risk of Hypercholesterolemia"

_cells, 2024, doi:10.3390/cells13060488_

Round 1

Reviewer 1 Report

Comments and Suggestions for Authors

Peer review Chaptal et al. round 1

In their paper, Chaptal et al. present two studies focusing on hypercholesterolemic (HC) and healthy subjects (HS). The first study examines the relationship between high LDL-C levels and the expression of adenosine A2A receptor (A2AR) and Flotillin-1, comparing HC matched patients to HS. The second, an experimental extension within the observational study, investigates the direct cellular response to high LDL-C levels by incubating PBMCs from HS in LDL-C enriched medium. This approach provides insight into whether cellular effects are directly due to increased LDL-C levels, thereby offering a controlled setting to study LDL-C's impact on cellular functions. The unique strength of this paper lies in using the PBMC incubation method to unravel these mechanisms.

Reviewer comments:

Abstract: Remove the underscore before the start of the third sentence. The abstract highlights the potential implications for understanding cholesterol toxicity and cardiovascular risks but could be improved if the authors discuss the direct clinical applications or how this research could be used in patient care or treatment strategies.

Introduction: Well written.

·        For clarity and consistency, consider revising "low-density lipoprotein-cholesterol (LDL-C)" to "low-density lipoprotein (LDL) cholesterol (LDL-C)" in your first mention. This will distinctly cover both the particle and its cholesterol content, aiding reader understanding in later references to LDL and LDL-C.

·        In the second sentence using "because" rather than "since" could be more effective for emphasizing causality. "Because" directly introduces a cause-and-effect relationship, making it clearer that the excess levels of cholesterol are the reason for the potential wider impact on tissue and organ functionality. "Since" can sometimes imply a time element or be used to provide additional information without as strong a causal emphasis as "because." Likewise in ” High LDL-C blood levels have been associated with an increased risk of lesion formation, while 55 high-density lipoprotein cholesterol (HDL-C) levels are generally considered protective, 56 since they are involved in the recycling and removal of excess cholesterol [2,3].”

·        Please spell out "PCSK9" initially to denote its full name, Proprotein Convertase Subtilisin/Kexin Type 9, for reader clarity. You can use the abbreviation "PCSK9" thereafter.

·        Add text describing the function of A2AR such as “it activates intracellular processes through G-proteins in response to adenosine binding” to the sentence in line 74: “Adenosine A2A receptor (A2AR) is a G-protein-coupled receptor expressed mainly in 74 immune cells and platelets, as well as in vascular smooth muscle and endothelial cells [12- 75 15].”

·        You need to add references for the following statement in line 83: “In recent years, a reciprocal link has been described between cholesterol metabolism and A2AR.”

·        Replace “subject” with “participant” throughout the document because your participants provided written informed consent and it emphasizes respect and autonomy, highlighting the voluntary and active role of individuals in a study, as opposed to "subjects" which might imply passivity. Secondly, it suggests a collaborative relationship between the researcher and the individual, acknowledging their crucial contribution, unlike "subject" which can indicate a one-directional interaction. It is a more person-centric term, recognizing the human aspect of those contributing to knowledge generation. Lastly, it aligns better with current ethical standards in research, which prioritize respect for individuals and their active participation.

Materials and methods

·        2.1 Human subject samples – I prefer "Study Population" over "Human Subject Samples" because it is a broader, more inclusive term that encompasses all participants in the study, reflecting both the hypercholesterolemic patients and the healthy subjects. It also aligns better with contemporary research language, emphasizing the collective group being studied rather than focusing on individuals as 'samples,' which can feel impersonal.

·        The term "interventional" in the study design might be misleading. It appears that while the core of the study is observational, focusing on the comparison of hypercholesterolemic and healthy subjects, there is an experimental component involving the incubation of PBMCs in an LDL-C enriched medium. This aspect adds an interventional dimension to the study. Therefore, the term "interventional" seems appropriate, as it reflects the experimental manipulation conducted to explore the cellular response to high LDL-C levels. This clarification should be included in the study's description to accurately convey its dual observational and experimental nature.

·        It's unusual for a study on hypercholesterolemia to exclude patients on hypolipidemic treatment, as these medications are common in managing high cholesterol levels. This criterion could limit the generalizability of the findings to the broader population with hypercholesterolemia using these medications to manage levels. It might be helpful if the rationale for this exclusion criterion was clarified, as it could impact the interpretation of the study's results.

·        Please ensure there is a space between the number and the unit in lines 121, 132, 154, 155, 175, 176, 211, 232, 245, and 250 for units like "mL," “mM,” and “kDa” to adhere to proper scientific notation. Additionally, I recommend reviewing the entire manuscript to check for any similar instances that might have been missed.

·        For proper scientific notation, replace the hyphen in "-80°C" with an En-dash, so it reads "−80°C". This notation is standard for indicating temperatures in scientific documents. Additionally, use En-dashes for ranges throughout the paper. Please review the entire manuscript to ensure this correction is applied consistently in all relevant instances.

·        I recommend removing the redundant phrase “Low Density Lipoprotein” following “LDL-” in section 2.3 because the abbreviation was explained in the introduction.

·        The best way to represent the chi-square statistic in writing is "χ²."

·        Remove the solid line at the end of line 167.

Results

·        Spell out "BMI" in line 172 i.e. body mass index and include the unit (e.g., kg/m²) after the numerical value in line 173. Also, ensure consistent spacing around the "±" symbol in all the measurements for readability (e.g., "BMI 23.89 ± 3.80" instead of "BMI 23.89±3.80").

·        In lines 172-173, the presentation of demographic data (percentages and ages) could be more consistent. For example, use a consistent format for presenting percentages (e.g., "71.0% women, 29.0% men").

·        Remove the term "triglycerides" in line 179 if its abbreviation has already been defined in the Introduction, to avoid redundancy.

·        Use the abbreviation "TC" instead of "total cholesterol" in line 180, since the abbreviation has been previously explained.

·        In line 183, rephrase “Lp(a) (lipoprotéine A)” to “lipoprotein A (Lp(a))”.

·        In line 186, use lowercase for "R" in "C-reactive protein (CRP)."

·        In lines 189-191, the description of differences in transferrin and ferritin levels could be clearer. For instance, stating "transferrin levels were lower and ferritin levels were higher in HC patients compared to HS" might be more straightforward.

·        In Table 1, modify the unit for BMI to the correct scientific notation: use "kg/m²" instead of "kg/m2."

·        In Table 1 and its footnote, since "CRP" and "Hcy" are already explained, remove their expanded forms from the table for conciseness. However, it's noted that "apoB" (Apolipoprotein B) and "Total-C" (which should be abbreviated as "TC" for Total Cholesterol) are listed in the table but not explained in the footnote. To maintain clarity and consistency, consider adding explanations for "apoB" and "TC" in the table footnote, matching the approach used for "CRP" and "Hcy."

·        The inconsistency in the spelling of "orosomucoid" in the text and "orosomucoïd" in the Table 1 should be addressed for consistency. The standard English spelling is "orosomucoid."

·        Address the interchangeable use of "p-value" and "p", along with inconsistent spacing in p-values in your manuscript. It's important to choose one term ("p-value" or "p"/ e.g., "p < 0.0001" vs. "p-value < 0.0001") and use it consistently throughout the document. Additionally, ensure that the spacing in p-value notations (e.g., "p < 0.0001" vs. "p=0.001") is uniform. In the description of statistical significance (lines 194-195), ensure consistent use of decimal points (use either commas or periods, but not both) and consistent space around the inequality symbols for p-values.

·        When comparing two groups, the use of the term "decreased" in the context of A2A receptor expression might be misleading. It would be more precise to say, "A2A receptor expression was found to be lower in HC patients compared to HS individuals." This phrasing more accurately represents a comparative result.

·        In section 3.4 the grammar in this section is mostly clear, but there's a minor issue in line 247 where "compared" is repeated. It should be revised for clarity and conciseness. The sentence could read: "...significantly over time in the presence of LDL-C compared to control."

Discussion: The discussion also successfully integrates previous research findings, providing a comprehensive analysis of the study's implications in the broader context of cholesterol metabolism and cardiovascular risk. However, the discussion could be enhanced by addressing the observed differences in ferritin and transferrin levels between HC and HS. Exploring these variations could offer valuable insights into the metabolic implications and potential cardiovascular consequences in HC patients.

·        In line 284, please spell out 'cAMP' for clarity.

·        Consider replacing 'since' with 'because' in line 291 to strengthen causal emphasis."

Conclusion:

In the Conclusion, a more detailed explanation on the potential clinical applications of these findings would enhance its impact. While you mention the influence of LDL-C on cellular components and the potential use of protein expressions as biomarkers, a clearer outline of how these insights could be translated into therapeutic strategies or diagnostic tools would be valuable. Elaborating on specific ways these molecular mechanisms might inform new treatments or early detection methods in hypercholesterolemia could offer practical implications for personalized medicine. This would not only contextualize the research but also provide a roadmap for future clinical applications.

Please note that the section detailing author contributions appears incomplete.

Comments on the Quality of English Language

Well written.

Reviewer 2 Report

Comments and Suggestions for Authors

Comments on the Quality of English Language

The quality of English language is fine.

Reviewer 3 Report

Comments and Suggestions for Authors

The authors analyzed the relationship between elevated blood LDL-C levels, intracellular cholesterol levels, and A2AR expression in a larger cohort of healthy subjects and untreated hypercholesterolemic patients. They found that A2AR and Flotillin-1 expression is lower in HC patients 26 than in HS and negatively correlated to LDL-C levels. The authors claim that LDL-C affects A2AR expression by impacting cholesterol-enriched membrane microdomains.

I have several comments:

Abstract:

-         Line 20. Why However? I do not understand.

-         Lines 23 and 24. Why did the authors choose to analyze A2AR and flotillin-1? There is no clear explanation of the reason for the choice.

-         Overall, the abstract sounds confusing and should be revised and rewritten. What does it mean a positive correlation between A2AR and flotillin-1? It is not clear.

Text

-         Line 109: I would not consider “normal” an LDL-C level of 160 mg/dL

-         Suppl figure 1 is not necessary. It is quite expected that apoB and /or LDL-C are directly proportional to LDL-C or TC, respectively.

-         Figure 1. Please, provide a better picture of the gel and also an uncropped version of the same.

-         Did the authors evaluate if there is a correlation between the reduction of A2AR and flotillin-1 in PBMC incubated with LDL-C enriched medium?

-         The main objection, however, is the lack of a mechanism. How does LDL-C induce A2AR and flotillin-1 expression changes? Is it causing the degradation of A2AR or it reduces its synthesis? The authors postulate a change in membrane microdomains, but they do not provide any proof.

-  Did the authors measure the expression of inflammatory markers or evaluate the cholesterol-enriched membrane domains? 

Round 2

Reviewer 3 Report

Comments and Suggestions for Authors

Text -         Line 109: I would not consider “normal” an LDL-C level of 160 mg/dL This value has been defined as normal accordingly to the recommendations of the “Haute Autorité de Santé” (HAS) “Major dyslipidemia: management strategies- recommendation for good practice”. Published online Dec 06 2018, and of VIDAL (www.vidal.fr/maladies/recommandations/dyslipidemies-1469.htlm#prise-en-charge).

               The guidelines the authors are referring to are quite old. There are more recent updated European and American guidelines indicating that an LDL-C level of 160 mg/dL is NOT normal. The authors should consider this.

-         Did the authors evaluate if there is a correlation between the reduction of A2AR and flotillin-1 in PBMC incubated with LDL-C enriched medium?  We have performed the experiments three times and for a correlation study a sample size of 3 is not sufficient to have a correct degree of correlation. However, as shown in Figure 4, we observed a concomitant decrease of A2AR and Flotillin-1 expression already after 6h incubation, which suggests a correlation.  -      

The authors decided on the experimental approach and stated in the abstract that “A2AR and Flotillin-1 proteins levels were positively correlated.” This is not a suggestion but a statement. Therefore, they should modify the abstract accordingly. In addition, the s in proteins should be deleted.

The main objection, however, is the lack of a mechanism. How does LDL-C induce A2AR and flotillin-1 expression changes? Is it causing the degradation of A2AR or it reduces its synthesis? The authors postulate a change in membrane microdomains, but they do not provide any proof.  -  Did the authors measure the expression of inflammatory markers or evaluate the cholesterol-enriched membrane domains?  We agree with the referee about the relevance and interest to understand the molecular and cellular mechanisms underlying LDL-C modulation of A2AR and Flotillin-1 expression. Nonetheless, as mentioned above, we believe that to address this question we need to perform a larger and complex investigation, which will be on his own the subject of another article.

I still believe that at least a hint on the possible mechanism should be researched. The measurement of a few inflammatory markers should be quite easy to accomplish.

Author Response

Point-by-point response to the referees’ comments. 

All the modifications have been highlighted in the manuscript text.       

Text -         Line 109: I would not consider “normal” an LDL-C level of 160 mg/dL This value has been defined as normal accordingly to the recommendations of the “Haute Autorité de Santé” (HAS) “Major dyslipidemia: management strategies- recommendation for good practice”. Published online Dec 06 2018, and of VIDAL (www.vidal.fr/maladies/recommandations/dyslipidemies-1469.htlm#prise-en-charge).

               “The guidelines the authors are referring to are quite old. There are more recent updated European and American guidelines indicating that an LDL-C level of 160 mg/dL is NOT normal. The authors should consider this.”

We removed the term “normal” from the sentence. (page 8, line 135).

-         Did the authors evaluate if there is a correlation between the reduction of A2AR and flotillin-1 in PBMC incubated with LDL-C enriched medium?  We have performed the experiments three times and for a correlation study a sample size of 3 is not sufficient to have a correct degree of correlation. However, as shown in Figure 4, we observed a concomitant decrease of A2AR and Flotillin-1 expression already after 6h incubation, which suggests a correlation.  -     

“The authors decided on the experimental approach and stated in the abstract that “A2AR and Flotillin-1 proteins levels were positively correlated.” This is not a suggestion but a statement. Therefore, they should modify the abstract accordingly. In addition, the s in proteins should be deleted.”

We referred to the positive correlation found in the whole participants population and described in Figure 3. To avoid any confusion, we modified the sentence in the abstract “A2AR and Flotillin-1 proteins levels were positively correlated in the whole study population.”. (page 2, line 36).

The main objection, however, is the lack of a mechanism. How does LDL-C induce A2AR and flotillin-1 expression changes? Is it causing the degradation of A2AR or it reduces its synthesis? The authors postulate a change in membrane microdomains, but they do not provide any proof.  -  Did the authors measure the expression of inflammatory markers or evaluate the cholesterol-enriched membrane domains?  We agree with the referee about the relevance and interest to understand the molecular and cellular mechanisms underlying LDL-C modulation of A2AR and Flotillin-1 expression. Nonetheless, as mentioned above, we believe that to address this question we need to perform a larger and complex investigation, which will be on his own the subject of another article.

“I still believe that at least a hint on the possible mechanism should be researched. The measurement of a few inflammatory markers should be quite easy to accomplish.”

The whole plasma collected from patients and healthy subjects has been completely used to quantify all the general biological markers of inflammation (see Table 1). We tried to assess by ELISA TNF-a (BD) and Interferon g (Invitrogen) levels in plasma obtained after Ficoll gradient. No difference was detected between 5 patients and 5 healthy participants. However, the levels were below the assay sensitivity (1.7 pg/ml for TNF-a and 1.0 pg/ml for INF-g). If required by the editor in chief, these results will be included in the manuscript.